# Effect of Relative Humidity on Transfer of Aerosol-Deposited Artificial and Human Saliva from Surfaces to Artificial Finger-Pads

**DOI:** 10.3390/v14051048

**Published:** 2022-05-15

**Authors:** Maurice D. Walker, Jack C. Vincent, Lee Benson, Corinne A. Stone, Guy Harris, Rachael E. Ambler, Pat Watts, Tom Slatter, Martín López-García, Marco-Felipe King, Catherine J. Noakes, Richard J. Thomas

**Affiliations:** 1Defence Science Technology Laboratory, Porton Down, Salisbury SP4 0JQ, UK; mdwalker1@dstl.gov.uk (M.D.W.); jcvincent@dstl.gov.uk (J.C.V.); cstone@dstl.gov.uk (C.A.S.); gharris1@dstl.gov.uk (G.H.); rambler@dstl.gov.uk (R.E.A.); pwatts@dstl.gov.uk (P.W.); 2School of Civil Engineering, University of Leeds, Woodhouse Lane, Leeds LS2 9JT, UK; l.benson@leeds.ac.uk (L.B.); m.f.king@leeds.ac.uk (M.-F.K.); c.j.noakes@leeds.ac.uk (C.J.N.); 3Department of Mechanical Engineering, University of Sheffield, Mappin Street, Sheffield S1 3JD, UK; tom.slatter@sheffield.ac.uk; 4Department of Applied Mathematics, School of Mathematics, University of Leeds, Woodhouse Lane, Leeds LS2 9JT, UK; m.lopezgarcia@leeds.ac.uk

**Keywords:** touch, transfer, surface, saliva, respiratory, gastrointestinal, virus

## Abstract

Surface to hand transfer of viruses represents a potential mechanism for human exposure. An experimental process for evaluating the touch transfer of aerosol-deposited material is described based on controlling surface, tribological, and soft matter components of the transfer process. A range of high-touch surfaces were evaluated. Under standardized touch parameters (15 N, 1 s), relative humidity (RH) of the atmosphere around the contact transfer event significantly influenced transfer of material to the finger-pad. At RH < 40%, transfer from all surfaces was <10%. Transfer efficiency increased markedly as RH increased, reaching a maximum of approximately 50%. The quantity of material transferred at specific RHs above 40% was also dependent on roughness of the surface material and the properties of the aerosol-deposited material. Smooth surfaces, such as melamine and stainless steel, generated higher transfer efficiencies compared to those with textured roughness, such as ABS pinseal and KYDEX^®^ plastics. Pooled human saliva was transferred at a lower rate compared to artificial saliva, indicating the role of rheological properties. The artificial saliva data were modeled by non-linear regression and the impact of environmental humidity and temperature were evaluated within a Quantitative Microbial Risk Assessment model using SARS-CoV-2 as an example. This illustrated that the trade-off between transfer efficiency and virus survival may lead to the highest risks of fomite transmissions in indoor environments with higher humidity.

## 1. Introduction

Transmission of pathogens may involve a range of routes and has high complexity due to behavioral and situational circumstances of humans living in complex environments. One route that can be pertinent to both respiratory and gastrointestinal pathogens is transmission by touching contaminated items, such as high-touch fomites (e.g., door handle, table surfaces, arm rests, utensils, etc.), or contact with individuals (e.g., hand shaking) [1,2,3,4].

Predictive infection risk models are reliant on transfer efficiency (TE) data of microorganisms from a surface to a finger to predict likelihood of infection under specific situations. However, reported TE estimates exhibit significant variability due to a range of factors that influence transfer. Human volunteer studies will be naturally variable due to differences in finger-pad contact area, applied touch pressure, and presence of skin hydration through secretion of sebum and sweat [5,6,7]. Relative humidity (RH) and temperature that control drying state have been reported to be important [3,8,9,10,11]; however, in many studies RH is not reported or the RH range in which studies are conducted varies considerably and does not explore tight ranges. During daily activities, behavioral aspects, such as contact complexity (frictional aspects of grip and sliding touch motions, multiple touches, frequency, sequence of surface contact, transfer direction, and efficiency of transfer to facial mucosae), contribute to quantity transferred [12,13,14,15,16,17]. Surface properties of the fomites play a role, with factors such as roughness and porosity influencing transfer [3,10]; indeed, non-porous and porous surfaces differ markedly at both low and high RH, with porous surfaces giving transfer efficiencies less than 13.4% [3]. An added intricacy for microbes is time-dependent survival on surfaces that could be either the fomite or human skin (i.e., hands; [9,18,19,20,21]).

Zhao and Li (2021) identified that the large errors (ranging from 40 to 100% relative standard deviation, RSD) associated with TE data could be improved by better control of influencing factors and systematic experimentation. Indeed, a large improvement to ~35% RSD was observed, enabling identification of surface roughness, liquid drying state, and sequential touch as important factors in contact transfer. Other factors were less important, for example, touch force had a small effect (10–15%) across the range 2 to 30 N. The resultant mechanistic model developed by Zhao and Li (2021) presents a frame to explore the relative importance of parameters relevant to fomite transmission [8]. A limitation of current studies is that pathogens are likely formed and deposited as aerosol rather than as liquid droplets onto surfaces humans interact with frequently [4,22,23].

Quantitative Microbial Risk Assessment (QMRA) is a framework for identifying and quantifying the key risks of pathogen transfer [24]. A key component of QMRA is exposure assessment, which is concerned with determining a pathogen’s survival in the environment, the nature and frequency of exposure events, and the size of the received dose that results from such exposures. The transfer efficiency estimates from a surface to a finger are one component, and it is this aspect that is described here. The overarching aim of this study was to develop an experimental system that supports deposition of viruses onto surfaces in a controlled manner enabling linkage of deposition to onwards transfer onto artificial finger-pads (AFP). The controllability allows isolation and investigation of factors considered important for fomite transmission and touch transfer of materials in general and particulates such as viruses or bacteria residing within the bulk deposited material. Herein, we describe characterization of a novel system and the impact of environmental humidity, surface complexity, and liquid rheology on quantity of material transferred. The limitations of the approach and areas for improvement in the experimental system are also described. The experimental data are used within a QMRA model to make predictions about the transfer efficiency of SARS-CoV-2 from surfaces to human skin, and how this is affected by environmental conditions.

## 2. Materials and Methods

### 2.1. Coupon Materials

Coupons representing high-touch surface materials included two metals (stainless steel 304 and aluminum alloy 5251) and four plastics (melamine D/S laminate, acrylonitrile butadiene styrene [ABS], smooth and pinseal texture versions, and KYDEX^®^ haircell texture) (Stead & Wilkins Ltd., Dartford, UK). Materials are denoted SS304, ALU, MEL, ABSS, ABST, and KYD within the manuscript. Coupon dimensions were 50 × 25 × (0.9–1.5) mm, providing a surface area of 1250 mm^2^. The coupons varied in surface area (Appendix A), however, the observed variance of 2.24% did not significantly impact transfer efficiency calculations. Coupons were cleaned prior and between experiments to standardize coupon surface. Heavily scratched coupons were discarded. Initially, the surface of the coupon was wiped with a commercial cleaning product containing detergent (Lancare Multiclean, MTM Services Ltd., UK) to remove gross contamination (e.g., grease or residual fingerprints). The coupons were then rinsed in the detergent for 5 min, washed twice in warm water with mixing, followed by two washes in deionized water, and a final wipe with 70% (*v*/*v*) ethanol.

Surface analysis of the coupons was undertaken by digital microscope (40–1000× magnification; Jiusion, China). Roughness parameters were determined using a Surftest SJ-410 surface roughness meter (Mitutoyo, Andover, UK).

Determination of contactable area for SS304 and ABST coupons and human finger-pad was performed by 3D non-contact profilometry using focus variation technique by an InfiniteFocusSL (Bruker Alicona, Graz, Austria). Briefly, scans of the surface topography of the items were taken and constructed into a 3D image. The software was used to draw a projected area which was moved at intervals through the z-plane to a depth dependent on the items’ surface roughness until the entire surface area was visible (finger-pad and ABST, 200 µm, and SS304, 20 µm). Each z-slice produced an above surface volume which by dividing by the z-plane length was converted to above surface area. The percentage area above surface of a given z-slice was calculated by dividing the above surface area by the projected area. Further detail is provided in Appendix A.

### 2.2. Aerosol Deposition onto Coupons

The aerosol deposition system comprised a stainless steel 416 box (WDH: 450 × 450 × 450 mm; Appendix A) with a circular port at the top for positioning of the aerosol generator (25 kHz ultrasonic nozzle, Sono-Tek Corporation, Milton, NY, USA). Four removable SS416 trays contained recesses for placement of coupons. A total of 112 coupons per spray could be used with 28 per individual tray. Prior to deposition, coupon loaded trays were passed under an ionizing bar (Hyperion 971, Meech International, Whitny, UK) for 5 min to minimize coupon surface charge affecting the aerosol deposition. An internal 12 V fan (axial) operating at 6 V provided uniform mixing of air and aerosol and was positioned to minimize directional flows, aiding reproducibility of aerosol deposition. A digital microscope camera (0 to ×200 magnification; Teslong MS100, Irvine, CA, USA) provided visualization of the spray, enabling deployment of a sliding tray with absorbent cloth (PIG matting, New Pig Ltd., Glasgow, UK) to capture larger drips from the nozzle at the start and end of the spray. Spray fluid was fed by syringe at a rate of 300 µL·min^−1^ for 10 min (PHD Ultra syringe drive, Harvard Apparatus, Holliston, MA, USA) into the 25 kHz Sonotek nozzle set to 50% power level. Temperature and humidity were recorded by TinyTag View 2 (TV-4505; Gemini Data Loggers, Chichester, UK). Humidity within the box was maintained at 100% RH and 24 °C by passing air through a heated bubbler at ~5 L·min^−1^; the wet air was fed into the box via a condensation catch pot. Some experiments used pipetted 1 µL droplets deposited centrally on the coupon to represent gross liquid contamination of surfaces rather than aerosol deposits. Controls included buffer blanks for both aerosol and liquid deposited samples, and spiked buffer and coupon spiked for the liquid deposited samples.

An artificial saliva recipe was used as a surrogate respiratory fluid [25]. Pooled human donor saliva (LEE Biosolutions, Maryland Heights, MO, USA) was used to understand rheological differences. In both fluids, 1 g·L^−1^ sodium fluorescein (MERCK Life UK Science Ltd., Gillingham, UK) was added as a tracer. Total solids in artificial saliva were calculated as 6.82 g·L^−1^. Total solids in human saliva were derived from Khan et al. (2008). Unstimulated resting condition saliva was reported as 4.58 ± 0.36 g·L^−1^ total solids [26]. Density and refractive index measurements were taken using DensitoPro and Refracto 30PX, respectively (Mettler-Toledo Ltd., Leicester, UK). The protein concentration was measured using a Pierce BCA Protein Assay kit and Multiskan Ascent spectrophotometer (ThermoFisher Scientific, Basingstoke, UK). A Fisherbrand Hydrus 300 pH meter (Thermo Orion, ThermoFisher Scientific, Basingstoke, UK) was used to measure pH. Viscosity as a function of shear rate was measured using a Rheometrics RSF2 teststation (Gemini BV, Apeldoorn, Netherlands). Particle size measurements of aerosol were taken using a Welas^®^ digital 3000 spectrometer with two 2300 heads (range: 0.2–105 µm, 5 L·min^−1^; PALAS GmbH, Karlsruhe, Germany) that operates at 90° scattering angle using a high-intensity white light source.

### 2.3. Production and Characterization of Artificial Finger-Pads

The artificial finger-pads (AFP) were generated from Dragon Skin^TM^ FX-Pro silicone (DDS; 2A Shore hardness; Bentley Advanced Materials, Feltham, UK) via development of a master cast and mold. The process was then scaled for batch production of six AFPs at a time. A non-stabilized artificial eccrine sweat–sebum emulsion (Verulam Scientific Ltd., Haynes, UK) was applied to each finger-pad before use in contact transfer experiments. The production process is described in detail in the Appendix A.

Real contact area of AFP and human finger-pads was obtained by ink printing adapted from the method of Liu et al. (2018) [27]. Briefly, AFPs were pressed onto a fingerprint ink pad, then using a commercial force-rig (MultiTest 2.5-i, Mecmesin Ltd., Slinfold, UK), driven down onto paper to create an impression at 1 or 15 N force for 1 s. Human fingerprints were generated by the same process, except the paper was placed on a balance (model CB 12K1N, KERN & Sohn GmbH, Balingen-Frommern, Germany). The ink fingerprints were photographed and image processed (ImageJ, University of Wisconsin, Maddison, WI, USA) to calculate the effective contact areas (i.e., contact area of ridges).

### 2.4. Measurement of Surface Energy and Dynamic Contact Angles

Surface energy was measured by the sessile drop method using a KRÜSS DSA100 drop shape analyzer (KRÜSS Scientific, Hamburg, Germany). Due to the limited surface area for analysis, 1 µL droplets were used. The probe liquids were water, hexadecane, and ethylene glycol. The use of these liquids allowed polar and dispersive components of the surface energies of the samples to be determined, as well as the total surface energy. Calculations were made using the Owens, Wendt, Rabel, and Kaelble method using the DSA4 software supplied with the instrument from 3 independent assessments with 10 replicates.

Dynamic advancing (θ_Adv_) and receding (θ_Rec_) contact angles were measured using a KRÜSS K100 tensiometer running ADVANCE software (KRÜSS Scientific, Hamburg, Germany). The surface tension of the artificial saliva used in the experiments was measured by pendant drop analysis using a KRÜSS DSA100 drop shape analyzer (KRÜSS Scientific, Germany). Experiments were conducted at 20 ± 0.1 °C, the temperature being maintained by a thermostatic jacket controlled by a Peltier temperature control unit. The immersion vessel was 50 mm in diameter and the software determined the minimum fill volume based upon the vessel size and sample dimensions. ABSS, ABST, and KYD coupons possessed different textured sides. To control for any texture or edge effects, two coupons were thinned and glued back-to-back using either epoxy (KYD) or acetone (ABSS, ABST), so the two external sides were the sides used for contact experiments. Samples under test were immersed at a speed of measurement 5 mm/minute to a depth of 5 or 20 mm. Contact angles were then calculated in using the ADVANCE software from 5 independent measurements. The difference between θ_Rec_ of recipient finger-pad and donor coupon surfaces (Δθ_r_) was then calculated.

### 2.5. Determination of Transfer Efficiency from Coupon to Artificial Finger-Pad

A schematic of the experimental assay is shown in Appendix A. A humidified cloche produced from metal frame and plastic sheeting housed the deposition box and force-rig. Humidity was generated within the cloche using an ultrasonic fog generator (Exo Terra, Rolf C Hagen (UK) Ltd., Castleford, UK), and modulated using manual on/off control. Lower humidities were controlled in the same manner using dry compressed air injection. Transfers at various humidities were assessed (25 to 85%) within tight RH ranges (±2% RH tolerance measured within 50 mm of point of contact transfer). Average temperature during transfers was 22.9 ± 0.6 °C (range: 20.6 to 24.2 °C). AFPs and coupons were equilibrated within cloche environment for 30 min prior to conducting transfers. Prior to use, coupons were checked to ensure consistency of aerosol deposit (i.e., no visible large splashes or other non-uniformities). Liquid droplet transfers were assessed immediately after deposition to prevent evaporation effects. Temperature and humidity were monitored 2 cm from the contact event (TinyTag View 2). AFPs were magnetically mounted in an inverted position in a fixture attached to the moving load-cell of the Multitest 2.5i system, whilst coupons were placed onto an anvil mount and fitted with a vacuum line to prevent lifting due to stickiness/suction on AFP—coupon separation. Contact transfer events were conducted at 15 N for 1 s with a final approach velocity (10 mm/min), controlled by commercial software (Emperor^TM^ Force, Mecmesin, Slinfold, UK).

After the transfer event, both the coupon (donor) and AFP (recipient) were placed into separate 10 mL volumes of borate buffer solution (pH 9.2; Fisher Scientific, Loughborough, UK) and shaken for 5 min to recover fluorescein into solution. Fluorescence units (FU) for both recipient and donor were measured by spectrofluorometer using excitation and emission wavelengths of 435 and 500 nm, respectively (Perkin-Elmer LS55, Beaconsfield, UK).

Transfer efficiency (*TE*) was calculated according to Julian et al. (2010) expressed as a percentage [12]. Studies using deposited aerosol used a modified equation to account for the experimental determination of real contact area (λ) of the AFP at a given force and coupon surface area (θ). *F* and *C* represent recovery from recipient (i.e., finger) and donor (i.e., coupon) surfaces, respectively, expressed as either FU, mass, or mass per unit area. Note for liquid transfer a volume was used (1 µL) that did not spread from underneath the AFP, ensuring confidence that the entire contaminating fluid volume was contacted.
(1)$$TE\;liquid\;droplet=\frac{F}{F+C}\;\times 100$$
(2)$$TE\;aerosol=\frac{F}{\left(F+C\right)\times \left(\lambda /q\right)}\times 100$$

Transfer efficiency data were cleaned through assessment of individual datasets. Each experiment comprised six individual finger-pad/coupon interactions, and data were excluded if fluorescence of the coupon or AFP exhibited greater than 10% variation in the mean for the batch of AFPs.

### 2.6. Quantitative Microbial Risk Assessment (QMRA) Modeling

A statistical regression model that describes the relationship between absolute humidity (AH) and transfer efficiency for the five surface materials was developed to inform an exposure assessment of fomite transmission of SARS-CoV-2. The predictions from this model were fitted to measurements of transfer efficiency of artificial saliva and measurements of SARS-CoV-2 deactivation reported in the literature under different environmental conditions [28]. A detailed account of the model development, fitting, and selection procedure is provided in the Appendix A.

#### 2.6.1. Development of Linear Regression Model of Transfer Efficiency Data

Data were available from 487 measurements of transfer following contacts at 15 N pressure between AFPs and coupons of five materials: ABSS (*n* = 47), ABST (*n* = 53), KYD (*n* = 59), MEL (*n* = 175), and SS304 (*n* = 153). This was alongside measurements of the quantity of fluorescein recovered from the finger, *F*, and coupon, *C*. Each contact had records of the temperature (°C) and RH. In addition, there was a record of which of the ten batches of AFP were used in each contact experiment. The temperatures and relative humidities (RH) in the contact experiments were converted to absolute humidity (*AH*; g·m^3^), as small variations in temperature at a given RH lead to small, though non-negligible, variations in *AH*.

The coupon-to-finger transfer efficiency, *T* (expressed as a number between 0 and 1), is the proportion of fluorescein that is transferred to the finger from that part of the coupon with which the finger comes into contact. The family of beta distributions is a natural choice for modeling proportions. Thus, a non-linear regression model with a beta response variable, *T*, and *AH* (g·m^−3^) and coupon material as predictors was fitted according to Equation (3). The quantity μ_AH, material_ represents the mean transfer for a given *AH* and material (ABSS, ABST, KYD, MEL or SS304), and ϕ_material_ is the precision parameter for each material. A beta distributed quantity, *T*, with mean and precision *μ* and *ϕ* has variance
(3)$$var\left(T\right)=\frac{\mu \left(1-\mu \right)}{\phi +1}$$
so that a larger precision corresponds with a smaller degree of variability in the transfer efficiency at a given mean. In the contact experiments, the artificial finger comes into contact with only a proportion, A, of the coupon surface area (1250 mm^2^). Using data from a set of separate contact area measurements (see Appendix A), *A* was found to be distributed normally, with mean m_A_ = 0.1865 and standard deviation σ_A_ = 0.0224.
(4)$$T=\mu_{AH,material},\phi_{material}∼Beta\left(\mu_{AH,material},\phi_{material}\right)$$

If F and C denote the quantities of fluorescein recovered, post-contact, from both the AFP and the entire coupon area (in appropriate units), respectively, then Equation (5) defines the proportion of fluorescein on the entire coupon that is transferred to the finger (*Y*). Thus, it follows that Equations (6) and (7) relate the indirectly observed transfer efficiency, *T*, the contact area proportion, *A* = λ/θ, and *Y*.
(5)$$Y\;\overset{\mathrm{def}}{=}\;\frac{F}{F+C}\;;$$
(6)Y=T×A
(7)$$Y_{i}|T_{i}∼Normal\left(m_{A}T_{i},\sigma_{A}T_{i}\right)$$
(8)$$\mu_{AH,material}=\frac{M_{material}}{1+e^{-\eta }};\;\eta =\beta_{0}+\beta_{1}\times AH+\tau_{b}$$

Several models were fitted to the data (see Appendix A), varying in how *μ_AH, material_* is specified, and one was selected that provided the best fit with the fewest parameters. The models were quantified using the artificial saliva data. The selected model relates the mean transfer efficiency to *AH* and material by Equation (8). The parameters *β*_0_ and *β*_1_ (the intercept and slope terms for the regression model) are shared among the five coupon materials, and *M_material_* (one for each material) are the mean transfers that are approached for large *AH*. The τ_b_, b =1,…, 10, are random effects associated with each finger batch used in the experiments and are included in order to account for the apparent correlations between measurements of *Y* using fingers from the same batch. It is assumed that the τ_b_ are each independently normally distributed with zero mean and standard deviation, σ, which was also estimated from the data. Posterior summary statistics for all parameters and AFP batch effect analysis can be found in Appendix A.

#### 2.6.2. Analysis of Effect of Environmental Factors on Risk of Fomite Transmission

The ‘U-shaped’, non-monotonic relationship between SARS-CoV-2 survival on surfaces and RH, described by Morris et al. (2021), was used to model environmental decay with either very dry or very humid conditions, leading to slower deactivation of the virus on surfaces. Low temperatures were favorable for virus survival, with a median virus half-life of over 24 h at 10 °C [28]. As the melamine dataset contained the largest number of data points, it was used as the example case to estimate the concentration of viable virus transferred to a human finger due to a single touch of a melamine surface 1, 10, or 20 h after the surface was inoculated with SARS-CoV-2. The analysis investigated combinations of environmental conditions (40, 65, 85% RH; 10, 22, 27 °C) and takes into account how these affect both virus survival and transfer from surface to finger.

It is assumed that the evaporation phase of the biphasic viral decay pattern reported by Morris et al. (2021) has completed by time *t* = 0, so that all viral decay occurring from *t* = 0 onwards is in the quasi-equilibrium phase, which is exponential in character. This is appropriate since the sprayed coupons are allowed to equilibrate prior to commencing the contact experiments. It is assumed at time *t* = 0 there is a viable viral concentration of *C*_0_ = 500 PFU cm^−2^. The viable viral concentration on the surface at time *t* > 0 was calculated using Equation (9).
(9)$$C_{t}=C_{0}exp\left(-\gamma t\right)$$
(10)$$\gamma =\frac{log2}{t_{0.5}}$$
(11)$$F_{t}=T\times C_{t}$$

Exponential decay rates (γ) for the nine combinations of RH and temperature are derived by taking the median viral half-lives, *t*_0.5_, reported in Morris et al. (2021) according to Equation (10). The posterior median values for the parameters β_0_, β_1_ and M_MEL_ are taken from Appendix A in order to calculate the mean transfer efficiency, *μ_AH, material_*, according to Equation (8) (assuming τ = 0). Together with the posterior median value for the precision parameter, ϕ_MEL_, the beta distribution for the transfer efficiency, *T*, is obtained as in Equation (4). Finally, the viable viral concentration transferred to the finger at time *t*, (*F_t_*), is given by Equation (11).

It is noted that the transfer experiments described in Section 2.5 focus on the effects of RH only, with the temperature constrained roughly within a 19–24 °C interval. To carry out this QMRA analysis, we assumed that transfer efficiency, for a fixed AH, is not affected by temperature. It is recognized that the uncertainty in the above estimates comes solely from the beta distributed transfer efficiency predicted by the regression model with parameters fixed at their posterior medians. Similarly, no posterior uncertainty is taken into account for the pathogen deactivation rates taken from Morris et al. (2021).

### 2.7. Statistical Analysis

Statistical analysis was performed using Prism 8.01 (GraphPad, San Diego, CA, USA) and MiniTab 19 (Minitab Inc, Ferguson Township, PA, USA). A significance level of 0.05 was used in all assessments. An *n*-way Analysis of Variance (ANOVA) was used to test the hypothesis that coupon depositions were different. The ANOVA was followed by Tukey’s multiple component analysis post hoc test. The Kruskal–Wallis non-parametric test was used to compare unpaired samples and the Student’s *T*-test for paired samples to understand the significance of particular humidities between surfaces.

## 3. Results

### 3.1. Development of a Standardized Experimental Process for Aerosol Deposition onto and Recovery from Surfaces

#### 3.1.1. Characterization of Coupon Surfaces

A range of coupon materials were selected, representing common high-touch surfaces found in a range of environments humans interact with daily basis (Table 1). The size of coupons selected ensured the AFP completely contacted the contaminated surface accounting for maximal spread at 15 N force, ensuring the AFP did not overlap the coupon edge. Therefore, the AFP and coupon could subsequently be placed in separate falcon tubes for the recovery of deposited and transferred material. The coupon is only half immersed within the 10 mL volume of recovery fluid, thus immediate shaking to recover the sample was required. Assay was conducted within a couple of hours to prevent occasional etching of the coupon (especially prevalent for ALU surfaces). The surfaces have natural complexity in terms of surface roughness. MEL and ABSS were the smoothest surfaces, whilst ABST and KYD were the two roughest surfaces, related to their embossed pinseal and haircell textures, respectively (Figure 1). The wettability and adhesion characteristics of surfaces may affect transfer [29]. This was assessed by contact angle measurements using static and dynamic techniques to understand both the surface energy and dynamic contact angles using artificial saliva (Table 1; the dispersive and polar components are shown in Appendix A). Surface tension of artificial and human saliva was determined at 64.8 ± 1.5 and 58.4 ± 1.1 mN·m^−1^, respectively, and used in assessment of the contact angles. The total surface energy was similar across all surfaces, as was the dispersive component, ranging from 25.8–33.7 and 22.5–26.1 mN·m^−1^, respectively. The polar components showed differences between the materials, where the metals (SS304, ALU) were much lower than the plastics (ABSS, ABST, MEL, KYD), ranging from 0.4–1.3 and 2.9–10.7 mN·m^−1^, respectively. The greater the advancing contact angle (θ_Adv_), then the greater the hydrophobicity of a surface. The θ_Adv_ increased for the surfaces in the manner: ABSS/ABST > SS304 > KYD > ALU > MEL. Thus, MEL was the most hydrophobic of the surfaces. The difference (Δθ_r_) between the θ_Rec_ of the sebum-coated silicone and coupon surfaces ranged from 11.1 to 28.9° in the manner ALU/ABST > MEL > SS304/KYD > ABSS.

Recovery efficiency of the artificial saliva–fluorescein matrix from coupons and AFPs was assessed by spectrofluorometry. A borate buffer directly spiked with 1 µL of the matrix provided maximal recovery estimates (68,184 ± 1156 FU, 1.7% CV, *n* = 10). The variation observed indicates error associated with pipetting of matrix volume and buffer dilution. Coupons and AFPs were similarly spiked with matrix, except assessment was directly post-deposition (wet droplet) or after 2 h drying to equilibrium with atmosphere (dried; 50.2% RH, 19.9 °C). Recovery efficiency for AFPs was 103.70 ± 1.70 and 100.72 ± 1.66% for wet and dried droplets, respectively (*n* = 6). Similarly, no significant difference in recovery efficiency was observed for the coupons irrespective of whether the droplet was dried or the presence of surface complexity (Appendix A).

#### 3.1.2. Aerosol Deposition

Deposition of aerosol onto surfaces will have associated variation that must be minimized to ensure that each coupon contains similar quantities of deposited material irrespective of position within the trays of the deposition box. The refinement of the experimental system to provide variation in deposition across all coupons of <10% is described in further detail in the Appendix A. Figure 2 represents depositions for MEL and SS304 coupons, demonstrating the consistency of deposition of artificial saliva and human saliva across 112 individual coupons. For artificial saliva, a mean mass of 0.168 ± 0.018 mg was observed (min and max: 0.124 and 0.210 mg; 10.8% CV). No significant difference was observed between each half tray (*p* = 0.51). Human saliva depositions also showed good spatial consistency across coupons. Two separate sprays showed 62,828 ± 5355 and 57,106 ± 3468 FU with CVs of 5.6 and 6.4%, respectively. Human saliva sprays had greater estimated mass deposited, averaging 0.289 ± 0.023 mg (min and max: 0.217 and 0.351 mg; 8.0% CV). The protein concentrations of artificial and human saliva were 0.682 ± 0.047 and 1.06 ± 0.11 mg·mL^−1^, respectively, whilst the pH was 6.96 ± 0.08 and 7.95 ± 0.2, respectively (*n* = 5).

Particle size distributions of artificial saliva, human saliva, and distilled water were similar at 100% RH (Appendix A). The density of water and artificial and human saliva was 0.9979, 1.0019, and 1.0003 g·cm^3^, respectively. The refractive index of water and artificial and human saliva was 1.3327, 1.3342, and 1.3336, respectively. The volumetric median diameters (VMD, dV 0.5) were 38.67, 36.53, and 38.62 µm for artificial saliva, human saliva, and water (minus fluorescein), respectively, when the box was maintained at 100% RH. The spread of particle sizes within a distribution is represented by the geometric standard deviations (σ_g_), and these were 1.305, 1.237, and 1.199 for artificial saliva, human saliva, and distilled water, respectively, at 100% RH. Addition of 1 g·L^−1^ sodium fluorescein to artificial or human saliva resulted in an increase in VMD observed at 45.87 and 48.0 µm, respectively (σ_g_ = 1.324 and 1.387, respectively). The size of the initial wet aerosol particle diameter would not be expected to significantly change with this additional dissolved mass. The Welas spectrometer detects particles by light scattering. It is probable that the observed increases with the addition of fluorescein salt were due to differences in the refractive index and fluorescence properties confounding the instrument’s interpretation of the particle size bins to which aerosol particles should be assigned. When the RH was reduced to 30% at the start of the spray (increasing to 70% RH over 2.5 min spray period), the VMD decreased to 21.40 and 18.47 µm, respectively, as expected due to evaporation. The distributions became more polydisperse with σ_g_ of 1.522 and 1.539, respectively, reflecting the fluctuating conditions within the deposition box at lowered RH.

### 3.2. Development of Artificial Finger-Pad with Characteristics of Human Finger Touch

#### 3.2.1. Topological Detail and Effective Contact Area

The topological detail (ridges and furrows) and skin wettability properties of human finger-pads determine friction during touch and are important to replicate in the AFPs [5,6,7,30]. Figure 3 illustrates the topological quality of replicating a human finger-pad surface. The small size of the entire unit enables complete immersion in the buffer, and alongside dissolvability of the matrix facilitates complete recovery of deposited or transferred matrix. The compressed ridges constitute the area of interaction for a pressed finger-pad against a surface representing the ‘effective contact area’ [27]. Effective contact area was compared between a human finger and a replica counterpart with a heavy touch press force of 15 N, which presented areas of 292 ± 96.7 and 233.1 ± 27.9 mm^2^, respectively (CV = 33.2 and 12.0%; Figure 4a; *p* = 0.33). In contrast, a lower force representing a light touch (1 N) gave areas of 132.9 ± 12.7 and 49.5 ± 13.7 mm^2^ for human and replica finger-pads, respectively, which were significantly different (CV = 9.6 and 27.7%; *p* = 0.0001). Reproducibility within a batch of AFPs was consistent, as indicated by comparison of effective contact area at 1 and 15 N for six replica finger-pads within a batch (Figure 4b).

#### 3.2.2. Influence of Sweat and Sebum Finger-Pad Wettability on Transfer

An eccrine sweat–sebum emulsion was applied until surface wetting properties consistent with a human finger were observed after application of 1 µL matrix droplets (Figure 5a). Analysis of the effect of the emulsion application on surface energy of the AFP surface was conducted (Figure 5b; Table 1) with an increase from 28.2 ± 0.34 mN·m^−1^ (28.2 ± 0.31 and 0.1 ± 0.03 mN·m^−1^ for the dispersive and polar components) to 56.2 ± 0.72 mN·m^−1^ (14.0 ± 0.28 and 42.2 ± 0.44 mN·m^−1^ for the dispersive and polar components) in the absence and presence of the applied emulsion. The value obtained for the AFP with emulsion applied was close to the value obtained for forehead skin of 50.7 mN·m^−1^ [31] which is sebum-rich, analogous to the finger surface. Dynamic measurements were also taken for both θ_Adv_ and θ_Rec_ for artificial saliva and sweat/sebum-coated and naïve silicone stubs. The presence of sweat/sebum decreased the θ_Adv_ of the silicone from 82.6 to 68.9°, thus reducing the hydrophobicity of the surface. Altogether, the contact angle data support the observations in Figure 5a that application of eccrine perspiration–sebum emulsion more closely represents wetting properties of real human finger-pads.

The effect of sweat–sebum application was assessed for liquid droplets (Figure 5c). No significant difference was observed between coated and uncoated AFPs for SS304, ABSS, and KYD. However, a significant difference occurred for ABST and MEL. Noticeably, the variability was much greater for uncoated (21.24–29.05% CV) compared to coated finger-pads (2.75–5.12% CV) across all surfaces. All further transfer experiments were conducted with application of emulsion due to the reduced variability in the contact event and influence observed on transfer efficiencies. Using liquid droplets, ALU produced highly variable results: 17.2 ± 5.8, 71.4 ± 11.0, and 43.4 ± 7.5% in three separate sets of contact experiments. Immersion in the borate buffer (pH 9.2) caused visible etching of the coupons. It is known that high pH can cause increased corrosion rates and subsequent oxidation can lead to variation in oxide layer depth and morphology [32,33]. Thus, further experiments with ALU coupons were abandoned until methods can be developed to standardize the surface; however, it serves as an example of the complex factors that can contribute to transfer between two surfaces.

### 3.3. Transfer of Artificial and Human Saliva from Surfaces to Artificial Finger-Pads

#### 3.3.1. Relative Humidity and Surface Complexity Influence Transfer of Artificial and Human Saliva

The effect of humidity on transfer of aerosol-deposited artificial and real saliva matrices to AFPs was assessed (Figure 6). Relative humidity, at which the contact transfer event was conducted, was tightly controlled within an environmental cloche. Irrespective of surface material, fully liquid material (1 µL deposited) was transferred for artificial saliva at 50.2 ± 2.98%. However, differences were observed for aerosol-deposited artificial saliva. The transfer of material from all surfaces was <10% when RH was kept at 40% or below. Thereafter, a significant increase in transfer occurred with increased RH, with the peak transfer reached dependent on the surface material. The smoothest surfaces, MEL, ABSS, and SS304 produced the steepest curves, peaking at 55% RH with transfer efficiencies of 54.8 ± 2.3, 50.5 ± 3.7, and 43.5 ± 1.3%, respectively. KYD and ABST have discernible surface roughness (Figure 1, Table 1), and formed shallower curves with reduced transfer of material at intermediate humidity. For comparison, at 55% RH the transfer efficiencies were 30.3 ± 1.3 and 23.0 ± 2.3%, respectively.

It was considered that the differences in transfer efficiencies from the different surfaces were due to the available surface area that an AFP could contact. The hypothesis was investigated for SS304 and ABST. Non-contact 3D surface profilometry (focus variation type) was used to determine the minimum surface area that can be contacted at various z-slice depths of the obtained 3D datasets of SS304 and ABST (Appendix A). The minimum surface area available for contact was 55.7% for ABST at a z-slice depth of 180 µm. In contrast, the SS304 minimum surface area was 50.0% at a z-slice of 18 µm (1.2% for ABST at similar slice depth of 20 µm). Thus, comparing the ABST and SS304 values, there is an order of magnitude difference in the topology of the surface, indicating greater surface area available for a given translation in the z-axis (i.e., ‘into’ the surface) for SS304 compared to ABST. This supports observations that the high transfer efficiencies observed for SS304 (also MEL and ABSS) are due to relatively lower surface roughness, with most of the deposited artificial saliva available for an AFP to contact. In contrast, the much rougher ABST surface will still have the same quantity of artificial saliva deposited across the coupon, however, the majority is deposited within the deep pits of the pinseal texture and cannot be contacted by the AFP.

The dependency on RH of transfer to finger-pad was compared for two materials, MEL and SS304, using artificial and human saliva as the deposited matrix. The curves were different, highlighting the influence of rheological properties of the deposited material (Figure 6b). Human saliva is a complex fluid formed through biological secretion and exhibits shear thinning properties compared to artificial saliva, which may influence transfer (Figure 6c).

#### 3.3.2. Development of a Regression Model for Humidity and Impact in QMRA Model

The data from the contact experiments for aerosol deposition of artificial saliva experiments were used to develop a non-linear regression model with a beta response variable, T, and AH (g·m^−3^) and the coupon material as predictors, according to Equation (4). See Appendix A for summaries of the parameter posterior distributions. The observed and predicted ratios, Y, of the quantity of fluorescein found on the AFP after each contact experiment to the total amount recovered from both finger and the coupon were analyzed. Figure 7 demonstrates the goodness-of-fit of the model by graphically comparing the data, Y (indicated by crosses), with the posterior-predictive distribution for replicate data points, Ŷ. These are values of Y simulated from the fitted model with parameter values drawn from their posterior distribution. By comparing Y with the model’s predictions of these same quantities for each material and a range of AH, Figure 7 suggests that the model fits well to the data. Most data points fall within the 95% predictive interval, centered on the median predicted value (i.e., an interval bounded by the 2.5 and 97.5 percentiles of their posterior-predictive distribution).

Thus, the fitted model parameters can be used to predict transfer efficiencies at different levels of AH. Using SS304 as an example, the posterior median value is taken (Appendix A) for the random effect standard deviation, σ. Thus, the random effect size, *τ*, associated with each finger batch for SS304 is distributed according to Equation (12). Hence, for a batch effect of, e.g., *τ* = 0.15, taking posterior median values for the parameters β_0_, β_1_, M_SS304_, and ϕ_SS304_ (Appendix A), gives a value for the linear predictor, *η*, for stainless steel at an AH of 10 g·m^−3^ according to Equation (13). Therefore, the mean transfer is given by Equation (14) and the predicted transfer efficiency, *T*, for SS304 at an AH of 10 g·m^−3^ is distributed as Equation (15).
(12)τ ~ Normal(0,0.223); 
(13)η=−7.401+0.738×10+0.15=0.129
(14)$$\mu_{10,SS304}=\frac{0.450}{1+e^{-\eta }}=0.239;\;$$
(15)T∼Beta(0.239,27.981)

The model of transfer efficiency was used alongside SARS-CoV-2 decay rates from surfaces at a range of temperatures and RHs [28] to predict the risk from touching an MEL surface contaminated with 500 PFU cm^−2^ as a function of time post-contamination (Figure 8). Time post-contamination before contact occurred was delayed by 1, 10, or 20 h to represent a range of contact situations. According to the model predictions, a temperature of 10 °C and RH in the range of 40–65% (AH = 3.78–6.11 g·m^−3^, top row of Figure 8) together present a low risk of transferring infectious virus to human skin. This is because the long survival of the virus on surfaces under these conditions is counteracted by the low level of transfer for such low AHs. However, the top-right panel suggests that a risk of fomite transmission at 10 °C emerges once the AH is sufficiently high to allow for the transfer of virus to skin in significant quantities.

Conversely, at the highest temperature, 27 °C, and for RHs of 40–85% (converting to 10.30–21.87 g·m^−3^ AH), there is initially a high degree of risk of transfer if the surface is contacted soon after inoculation, due to the high degree of transferability expected for large AH. However, this risk reduces with time between inoculation and exposure due to a higher rate of viral deactivation at 27 °C than at 10 °C. Referring to the middle row of Figure 8, at 22 °C, given enough time for the differential effect of RH upon virus survival to become apparent, the non-monotonic, U-shaped relationship between RH and viral survival is reflected in the risk of exposure; the risk is markedly lower at 65% RH than it is at 85% RH and falls away more rapidly with time than that at 40% RH. According to our model, at 22 °C and 85% RH (AH = 16.50 g·m^−3^) the transfer efficiency of virus from surface to skin is predicted to be close to the maximum transfer efficiency of 0.574 for melamine. Compounding this high level of transfer is a slow rate of viral deactivation found in highly humid environments [28].

## 4. Discussion

Virus transfer from fomite surfaces to finger-pads and onward transfer is a known transmission route for many viruses such as norovirus, but also in the event of a novel epidemic or pandemic virus outbreak, must be considered as one of the potential routes of transmission [4]. Investigating factors important for surface transmission of pathogens has been hindered by large variation in data. Variability in the TE data can be reduced by controlling and standardizing individual components within experimentation [8]. Adopting this experimental sensitivity analysis approach, a capability is described enabling analysis of soft matter, tribological, environmental, and microbiological factors associated with contact transfer. The experimental system comprised three components: (i) an apparatus enabling reproducible aerosol deposition onto coupons, (ii) the AFP replicating properties of the human finger including consistent contact areas, and (iii) methods to replicate and assay touch transfer events. The benefits of the new system as well as limitations and areas for improvement are discussed.

A significant issue in assessing TE of viruses within human deposited material is choosing the optimal deposit form. It is simple to produce and deposit small aerosol (<10 µm) or pipette 1 µL droplets, but due to inertia effects aerosol droplets bigger than about 10 µm become difficult to control and deposit in a reproducible manner. In addition, at some point in bulk studies (i.e., dried pipetted droplets), increased depth of dried material will cause the method used to fail due to the fact that only the top surface of any deposit can be touched. This is not the case for thinly deposited aerosolized material, where initial work undertaken in planning demonstrated that in bulk pipetted deposits, surface material would ‘shield’ deeper layers, resulting in a much lower calculated TE. In this study, the aerosol particle size generated was approximately 40 µm, to represent larger aerosol sizes generated during human cough events [22,34]. These sizes would readily deposit onto high-touch surfaces such as a keypads or table surfaces within the vicinity of an infectious individual, effectively forming a dried thin layer of single particles that represent what would be contacted by a finger-pad [23]. No attempt was made to achieve human-derived spatial variability in particle deposition as observed in the simulated cough experiments of Wang et al. (2021), but it is recognized this would occur in natural situations. Rather, control and reproducibility of deposition was a focus to reduce variation to <10% in order to homogenize the concentration under the contact area of the AFP. Humidity within the deposition apparatus was retained at 100% to ensure evaporation of particles did not occur during the deposition process (10 min) and that aging effects or evaporation kinetics of all deposited aerosol would be similar. Thus, analysis was limited to microscale interactions between saliva particles of known size and dried state and the selected surfaces and their properties. Good reproducibility of both initial aerosol distribution and deposition across coupons was observed with spatial variation <8% for the contact transfer experiments described. The artificial and human saliva aerosols were polydisperse (σ_g_ < 1.3, while monodispersity is defined as σ_g_ ≤ 1.1). Any drying effects would only occur once the coupons were removed from the deposition box into a humidity-controlled environment and would therefore be as uniform as possible across all the coupons at a particular RH that contact transfers were conducted.

Human finger-pads have topological complexity and lubrication through sweat and sebum that impact frictional and other surface properties [5,6,7,30]. Both aspects were represented in the AFPs. Application of eccrine sweat–sebum emulsion affected transfer of liquid droplets from MEL and ABST but also reduced variability within datasets. The presence of the lubricating layer improves contact between the AFP and the surface alongside altering the hydrophilicity of the AFP surface closer to a human finger. As a human finger presses onto a surface with increasing force, the finger-pad spreads and the ridges represent the finger-pad surface that contacts the opposing surface. This is defined as the effective contact area (ECA) and is an important aspect to represent and understand variation in the AFPs [27]. In this study, to minimize variation due to elastic deformity of the finger-pad during a contact, the AFP was constructed with a flattened surface. The measured ECA at 1 and 15 N of the volunteer finger in this study is consistent with values obtained during a review of contact properties of human fingers [7]. However, a difference in ECA at 1 N contact forces was observed between the volunteer and AFPs (Figure 5). This is likely due to differences in material viscoelastic properties and selection of the flattened AFP approach where the rapid deformation at lower forces below ~2 N in human finger-pads will not be represented [27]. A human finger is a composite of tissues, with each layer differing in depth and elastic properties alongside inter-subject variability [5,6,7,30]. In contrast, the AFP is composed purely of silicone (2A shore hardness). The elastic deformation (i.e., Young’s modulus) of a human finger-pad has been measured at 0.07–0.2 MPa for forces of 0.5 to 10 N [35] and ranging from 0.9 to 4 MPa depending on the subject [36]. Further development of the AFP could be undertaken to more closely control the application of lubricants and better represent the viscoelasticity of a human finger, e.g., [37]. Variation in quantity of lubricant applied may explain variation in TE data, as it is likely to vary between AFPs, AFP batches, and across the surface of individual AFPs. Measurement of the uniformity of sebum/sweat coated onto the AFPs has not yet been undertaken.

Transfer efficiency of both artificial and human saliva from surface to AFPs was highly dependent on RH across the range of 40 to 65% (Figure 6). The low TEs observed for both matrices below 40% RH (0.53–7.5% TE) match well to mean values of <0.2–21.7% for bacteria (*Escherichia coli*, *Staphylococcus aureus* and *Bacillus thuringiensis*) and bacteriophage (MS2) deposited in media and transferred from surface (glass, ceramic tile, laminate, stainless steel or granite) to human finger at 15–32% RH [3]. Acrylic did give higher TE and variability for all microorganisms except *S. aureus* [3], perhaps highlighting differences in the surface and that species-specific properties should be considered. For the same surfaces, much greater transfer of microorganisms occurred at RH 40–65%. The mean TE values for all microorganisms deposited in culture media from stainless steel and laminate were 37.4–57.0% and 27.4–63.5%, respectively (ranged from 19.5–99.0% and 1.9–100%, respectively) [3]. The variability can be explained by our study, where this RH range is associated with rapid changes in the drying state of the deposited matrix, as described in more detail below. Further, temperature differences between studies makes direct comparison difficult, as RH is a function of temperature. Therefore, absolute humidity should be reported in addition to or instead of temperature. Despite the variability, the TEs observed in this study across the same wide RH range are not too dissimilar. SS304 and MEL for both matrices ranged from 11.6–40.0% and 18.5–59.5%, respectively (Figure 6a,b). This perhaps implies that the bulk transfer of deposited matrix dominates TE rather than properties of the microorganism that reside within; further studies are required to understand this complex relationship and it is recognized that it will be influenced by the use of human compared to AFPs and differences in rheological and surface properties.

Interestingly, the point at which an increase in TE was observed for all surfaces was above 40% RH. The artificial saliva recipe used in this study was reported to undergo a phase change at water activities below 0.45 (i.e., 45% RH) to crystalline, amorphous, or mixed phase morphology in aerosol phase [38]; above this point water vapor in the atmosphere would support deliquescence of the artificial saliva. It is likely that the hygroscopicity of the deposited material is also influencing the transfer process to AFPs and might account for the difference in TE curves observed for SS304 and MEL between artificial and human saliva (Figure 6b). However, the differences are not fully understood. Both surfaces for human saliva show similar curves to artificial saliva, albeit with a lower TE. It appears that the human saliva responds differently to rehydration, though both are tacky at higher RHs. Human saliva may have a less pronounced phase change. Efforts were made to use an artificial saliva representative of human saliva in terms of salt and mucin composition and concentrations [25,39,40,41]. However, visual and rheological examination highlight that the two fluids are different (Appendix A and Figure 6c). Artificial saliva contains porcine stomach mucins mixed into solution, whereas the human saliva contains biologically secreted saliva mucins (i.e., MUC5B and MUC7), with difference glycosylation patterns [42,43]. These dissimilarities influence rheological properties and possibly water retention, perhaps accounting for the reduced TE profiles observed for human saliva compared to artificial saliva. It should be noted that human saliva is highly complex, changing through the course of a day and with factors such as nutrition, age, sex, and illness, which may alter composition, ratio of mucins, and rheological properties [39,40,41].

The differences between TE-deposited saliva and AFP (Figure 6a) are also influenced by the properties of the coupon surface. Textured coupons such as ABS pinseal and KYDEX^®^ haircell produced shallower curves compared to less textured surfaces such as melamine and non-textured ABS for the aerosol-deposited material (25 to 85% RH). Surface roughness, surface energy, and contact angle measurements were made to understand the relative role parameters specific to surface types may play. Contact angle measurements describe the wettability and adhesion characteristics of a surface. Thus, if an aerosol particle or droplet deposited, how well would it spread after initial deposition and during humidification changes occurring in natural environments (i.e., bathroom, kitchen surfaces during activity introducing heat and moisture into environment)? The quantity of liquid material transferred between two surfaces has been related to the difference between θ_Rec_, as they interact dynamically, and the capillary bridge eventually breaks during separation [29]. In this study, although differences in the polar component of surface energy and Δθ_r_ were observed between coupon materials, surface energy could not be related to differences in the TE profiles for the respective materials at present. This may be because other factors such as roughness and humidity dominate transfer. Indeed, roughness has been shown to influence contact angle measurements [44,45]. Additionally, the soluble nature of some of the components in the sebum/sweat mix would be expected to modify the advancing but not the receding contact angle data, as once liquid (or highly hydrated material) contacted the sebum/sweat these components would migrate into the mix.

The reason for which surface roughness influenced transfer was considered to be the availability of fractional surface area available for contact by the AFP, which will be lower for rough surfaces such as ABST and KYD compared to smoother, less complex surfaces (ABSS, MEL and SS). This was confirmed using the focus variation technique to determine the minimum available contact area for SS304 and ABST. At similar z-slice depths of 18–20 µm, the minimum surface area available for contact was 1.2% and 50% for ABST and SS304, respectively. The macroscale differences in roughness for ABST (and KYD) will support much of the deposited material being inaccessible to the AFP, and thus account for reduced transfer. The difference in TE curves between the smoother surfaces can be explained by microscale texture. Although similar in roughness profile, SS has a much more complex surface texture with greater tendency to develop scratches (Figure 1a,b,h) compared to MEL and ABSS (Figure 1d,f,i), with which the saliva aerosol will deposit and interact. These textures are not of the scale of ABST and KYD and thus the available area for contact more closely resembles that of a smooth surface. The TE of fully liquid droplets for all surfaces reached ~50% and thus was independent of roughness for the materials in this study (Figure 6a). It could be seen during contacts that the liquid droplet would move and spread under the influence of the AFP fingerprint micro-channels during the compression, then be drawn back by surface tension forces as the AFP was removed, sometimes leaving multiple small droplets where individual capillary bridges broke. In the cases of ABST and KYD, the deposited liquid sits proud on the surface peaks, thus the entire liquid volume is contactable in the same manner for smoother surfaces. It may be that surfaces with chemistry that significantly alter hydrophobicity and polarity behave differently. Behzadinasab et al. (2021) attributed the greater transfer of SARS-CoV-2 to artificial skin (Vitro-Skin^®^) to hydrophobicity in the case of Teflon compared to glass or stainless steel.

Similar observations that surface characteristics (material, roughness, porosity) and humidity/drying state influence transfer for microorganisms have been previously reported (see Table 1 in Zhao and Li, 2021; [3,8,10,11,17,46,47]). Three recent articles evaluated transfer efficiency of SARS-CoV-2 from surfaces to artificial finger systems developed from either Vitro-Skin^®^ overlaying either a real human finger or a polydimethylsiloxane support, or nitrile gloves [9,10,11]. It should be noted these studies lack the topological detail of a finger-pad surface present in both this study and that of Lopez et al. (2013). Transfer efficiencies were obtained from a range of surfaces including stainless steel. Todt et al. (2021), for stainless steel discs with 10^6^ TCID_50_·_mL_^−1^ deposited and dried for 1 h at 32–43% RH, produced a TE of ~1–2%. This matches TEs for stainless steel in this study, which over the same RH range generated TEs of 0.27–8.78% and 1.42–4.46% for artificial and human saliva, respectively. In contrast, the study of Behzadinasab et al. (2021) reported transfer of ~3% for stainless steel after drying for 30 min at 60–70% RH. In this study, much greater transfers were observed, approaching 45% for stainless steel, and perhaps indicating differences in method; the topological ridges and grooves of the finger-pad alongside the presence of sweat/sebum may support greater removal of material in this study. This could be evaluated in the future using finger-pads that lack topological detail. Zhao and Li (2021) describe the difficulties in interpretation of data from unsystematic experimental protocols and the associated large RSDs that range from 35 to 100% in the literature.

Butot et al. (2022) investigated SARS-CoV-2 (6 log_10_ TCID_50_ in PBS with MgCl_2_ and CaCl_2_ added) transfer from surface to gloved fingers for food (lettuce, cooked ham, vegetarian meat alternative, VMA) and food packaging surfaces (plastic, cardboard) as a function of drying state (immediate wet, dried for 1 h at room temperature, frozen for 24 h at −20 °C). Although cumulative transfer from surface to finger and onwards to another finger was measured, for comparability to the data in this study, only the initial surface to finger transfer is discussed. Transfer was greater for wet surfaces than dried irrespective of material, which supports the observations from Figure 6a,b. Material type had an effect with transfer of wet deposited material occurring at 40.5%, 28.9%, and 11.3% for lettuce, ham, and VMA, respectively. Similarly, the transfer of wet deposit on plastic was greater than cardboard at 25.3% and 9.2%, respectively [11]. Related to the observations in this study, differences in surface properties may result in the observations (e.g., roughness, porosity, hydrophobicity, and wicking effects); however, detailed characterization of surface structure was not performed. Interestingly, frozen material did transfer at intermediate efficiency between wet and dry values, with plastic supporting transferal of significant quantities at 10.9% [11]. Rate of defrosting will influence transfer and it may be that plastic supported a more rapid rate than cardboard or VMA. It is notable that RH was not reported, although as the experiments were conducted in a Biosafety cabinet, this is likely to be in the range equivalent to inlet air from the laboratory. The influence of RH described in this study on transfer efficiency is likely a factor in the large errors observed in some published transfer data (Appendix A). Most of the studies lack sufficient information to understand how an individual contact event relates to RH or drying state. It is recommended given the high dependency of transfer on RH for the range of surfaces used in this study that RH should be controlled, monitored, and reported in greater detail, particularly from the vicinity of the touch event. Other factors undoubtedly play a role in variation in published transfer efficiency data such as decay rate, surface properties, volunteer differences, and sampling/ elution efficiency.

This study focused on understanding the mechanistic principles that govern transfer of the bulk material from surface to a topologically representative silicone finger-pad. An assumption was that the virus will track with the bulk material, i.e., artificial or human saliva. However, it does represent a limitation. It is recognized that particulates such as viruses or bacteria were not assessed, and effects such as adhesion of microbes or evaporation edge effects such as crystallization patterns, coffee-ring formation, or microbial density are not accounted, which may influence both survival and transfer of virus to a finger-pad [48,49,50]. Most studies on virus survival and transmission tend to use culture media, buffers, or artificial saliva recipes; however, data presented in Figure 6b demonstrate biological respiratory fluids can behave differently from standard laboratory media and buffers. This is supported by published data showing bacteriophages deposited in sprayed droplets of buffer, water, artificial respiratory fluids, and human saliva onto glass slides differed in survival kinetics and crystallization patterns as a function of RH [48,51]. Further, studies using beads (100 nm or 2 µm) or phi6 bacteriophage show that the drying kinetics and fluid composition influence location within the dried droplets; in human saliva the beads were homogeneously spread through the matrix [48,50,51]. Future studies will explore this aspect and the role of physical size through using beads and microorganisms using the described touch transfer system.

The contact event itself is a complex mechanistic interaction. The model developed by Zhao and Li (2021) can be expanded based on our observations. Firstly, the consistency and adhesiveness of the deposited material (i.e., saliva) for both the substrate and the AFP will be influenced by humidity (i.e., altering drying state of deposit) [48,49,50]. Once the AFP ridges start to interact with the saliva on the coupon surface there will be regional mixing of the salts, surfactants, and mucins present in the saliva with salts and oils present on the AFP ridges. Thus, the resultant physicochemical properties of the transferring matrix (that would contain pathogens if present) such as hygroscopicity and rheology may be altered from the two component materials in an RH-dependent manner. The quantity of mixing is likely dependent on available contact area governed by surface roughness, contact force, and rheology. It is envisaged that force and time will cause further spreading and mixing of materials, particularly at higher RH, possibly creating greater contact with the AFP grooves [5,7,30]. Once maximal pressure and time is reached, the AFP will withdraw and the rheology and adhesiveness of the mixed material (determined by RH) will create liquid bridges that gradually stretch and break at a distance from the separating surfaces, with some material retained on the surface and some leaving with the AFP [29,52]. The entire process across the contact area of an AFP–coupon contact is likely a stochastic process due to the variability in drying state, concentration, and depth of saliva, as well as AFP coating and ratio of mixing across coupon and AFP as they press together and then move apart. Further research is required to understand how human finger-pads behave in comparison to the AFPs and how particulates residing within the deposited saliva would transfer. Aspects of real fingers, such as surface moisture, temperature, and local humidity under the finger as it contacts and physiologically reacts to the surface and ambient conditions [53,54], may also influence quantity of material transferred. The mixing effect of a sebum/sweat mix with a moist saliva deposit is currently an unknown variable. Both materials are sticky at higher humidities and ‘fingerprints’ are clearly seen after a contact. Better control over sebum/sweat application, a weakness in the current capability, would allow this to be investigated.

Experimental TE data should be viewed from the context that these are controlled experiments and surfaces; in the world human interactions with surfaces will be more complex still. High-touch surfaces humans interact with will likely have altered surfaces through such interactions and temporal processes (dirt/grime/sweat/sebum accumulation and removal, corrosion, biofilm formation, scratch development, and cleaning regimes). These will alter interactions and adhesive forces at the microscale level between deposited material and the interacting finger-pad and environmental surfaces. The experimental system described starts to enable the role of these intricate interactions to be unraveled in a controlled manner alongside techniques to understand surface topology and adhesiveness.

Via the regression analysis, relating transfer efficiency to coupon material and AH, described in Section 2.6 and Section 3.3.2, we obtained an empirical model that is able to predict the extent of transfer at temperatures in the range of 19–24 °C and relative humidity in the range of 25–85% for the five surface materials. Such a model should serve as a useful input into QMRA analyses that seek to quantify the risk of fomite exposure to pathogens. The QMRA analysis takes into account both transfer efficiency and viral inactivation and the resultant trade-off between these two factors. Overall, there is a complex relationship between environmental conditions, time between surface inoculation and contact, and viable viral concentration transferred to the finger. There are risks posed due to virus deposited onto surfaces under certain conditions, and this risk varies in degree significantly, depending upon environmental conditions. The way that the time elapsed between inoculation of the surface and contact interacts with environment conditions suggests that, e.g., frequently contacted surfaces may pose a risk of fomite exposure regardless of environmental conditions, whereas the risk associated with less frequently contacted surfaces is dependent on a combination of both temperature and RH, with risks likely to be persisting longer in more humid environments and at temperatures commonly experienced in indoor settings. Such higher humidity conditions may be more likely to be found in bathroom environments, and several studies have shown that higher levels of contamination have been found in toilets and bathrooms [55]. Greatest risk would be anticipated to be the initial contact events prior to dilution effects, as subsequent touches redistribute the viral load.

It should be noted there are limitations to the data input into the model; artificial saliva was modeled, and it is clearly seen that transfer is lower in a more biologically representative respiratory fluid such as human saliva. Batch effects were observed within AFP batches with wide variation in percentile ranges (Figure 8). As the transfer is a physical process, if all variables could be controlled then the TE data should show single lines for each coupon material and not the wide variability seen. Currently, it is not fully understood where this variability stems from. There are uncertainties in the sebum/sweat deposit, and as yet no group has been able to visualize this or quantify it pre-contact. In addition, the RH control does produce pulses of high and low RH, and these pulses take some time to mix in with the bulk air in the cloche. This could cause localized temporal changes in RH around the AFP/coupon event, leading to differences in RH during the contact compared to that measured. It is known that selecting coupons with more uniform surfaces leads to lower variability; this, like the surface texture and bulk deposit issues, would be due to scratches making material unavailable for a contact but available for elution.

The decay data of Morris et al. (2021), although comprehensive, are generated from 50 µL pipetted droplets of virus in culture media onto polypropylene disks. This represents a different hazard more akin to large liquid deposits from a ‘runny nose’ onto surfaces either direct or via hand. Indeed, no data to date exist on aerosol-deposited SARS-CoV-2 on surfaces. However, the general conclusions are similar to other reports where 1 to 50 µL pipetted artificial saliva droplets containing SARS-CoV-2 did not show differences in decay rates for a range of surfaces [18]. Biryukov et al. (2020) observed quicker decay at higher temperature but also that decay decreased linearly, decreasing RH. This may be a reflection of using artificial saliva rather than culture media. Despite this, it should be observed that deposited aerosol will represent a very different environment. The 40 µm^3^ aerosol particles generated in this study represent a volume of 0.04 picoliters. This is many orders of magnitude smaller than 1 to 50 µL droplets and it would be expected that the kinetics and interactions with surface and air as such tiny volumes dry will be very different. Studies exploring the breadth of volume sizes associated with respiratory-deposited material and decay rates in biologically relevant matrices could be useful for defining risk from surfaces.

## 5. Conclusions

In conclusion, a touch transfer system provided the capacity to uniquely control and investigate tribological, soft matter, and environmental and microbiological factors associated with surface to finger transfer from aerosol-deposited material. At the microscale level, this controlled mechanistic approach identified the interplay between environmental humidity, surface roughness, and fluid properties in defining the adhesiveness of the respiratory matrix for particular surfaces and its propensity to be removed via a touch event. Coupling transfer data with virus inactivation through a QMRA model allows the trade-off in environmental factors influencing exposure risk to be explored. Key conclusions from the study are:Transfer efficiency increases with increasing humidity for all materials studied. At RH <40%, transfer from all surfaces was <10%, and increased markedly as RH increased, reaching a maximum of approximately 50%.The quantity of material transferred at specific RHs above 40% was dependent on the roughness of the surface material and the properties of the aerosol-deposited material. Smooth surfaces, such as melamine and stainless steel, generated higher transfer efficiencies compared to those with textured roughness, such as ABS pinseal and KYDEX^®^ plastics.Pooled human saliva was transferred at a lower rate compared to artificial saliva, indicating the role of rheological properties.QMRA analysis using SARS-CoV-2 suggests that the highest risks of fomite transmission are likely in indoor environments with normal temperatures (around 22 °C) and higher humidity (>65% RH). This suggests humid spaces could pose a higher risk.

Future work will look to further support QMRA models through expansion of the dataset to include tribological (touch force, time, and repetition) and microbiological factors (particulate size, density i.e., virus, spore, bacterium) and structured investigation of material properties of surface, deposited matrix, and AFP (roughness, porosity, hydrophobicity, surface chemistry, particle size, viscosity, sebum depth). Finally, understanding the survival of virus deposited in relevant material in a representative manner would support greater interpretation in models (i.e., human respiratory fluids as aerosol or drips).

## Figures and Tables

**Figure 1 viruses-14-01048-f001:**
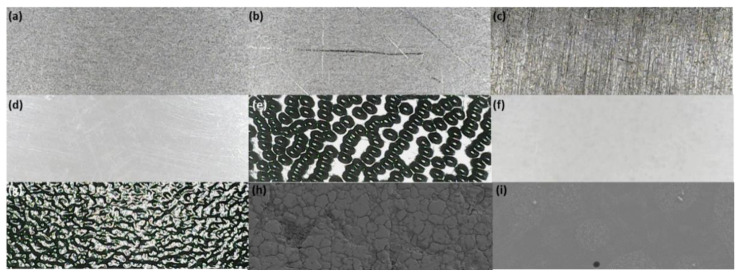
Microscopic analysis of coupon material surface textures. Light microscopy at ×20 magnification: (**a**) unscratched SS304, (**b**) scratched SS304, (**c**) aluminum 5251 alloy, (**d**) ABS smooth surface, (**e**) ABS pinseal texture, (**f**) melamine laminate, (**g**) KYDEX^®^ haircell texture, and SEM microscopy at ×1000 magnification. (**h**) SS304 showing surface complexity (×1000 magnification) and (**i**) melamine laminate showing deposited droplets of artificial saliva–fluorescein matrix and crystal formation (×250 magnification).

**Figure 2 viruses-14-01048-f002:**
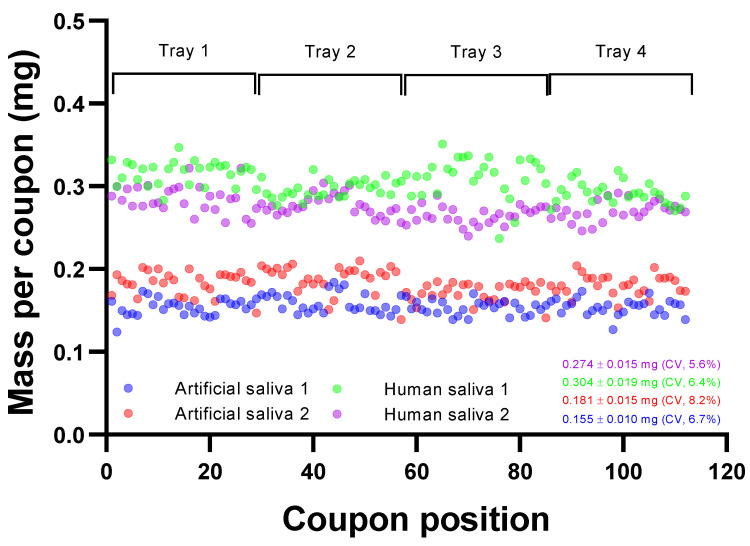
Deposition reproducibility on coupons as a function of position within aerosol deposition box. Artificial saliva 1 is MEL only; artificial saliva 2, human saliva 1 and 2 are mixtures of MEL and SS304. Fluorescence values are converted to mass through knowing the mass in g·L^−1^ of total solids in the spray suspension and the fluorescence of the initial spray suspension.

**Figure 3 viruses-14-01048-f003:**
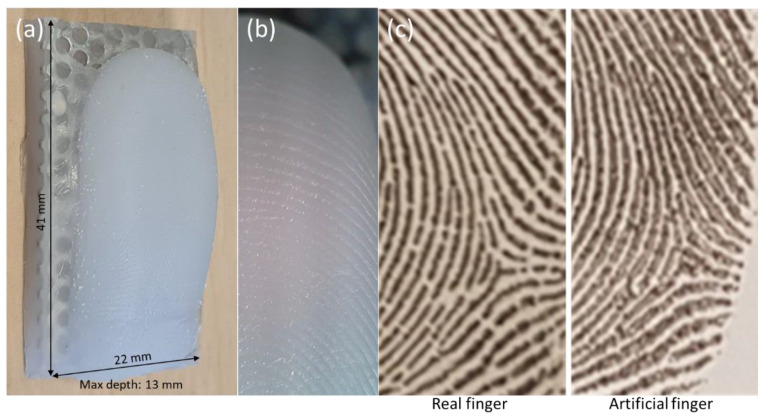
Topological quality of artificial finger-pad. (**a**) Entire artificial finger-pad, (**b**) magnified view of portion showing topology (×20), and (**c**) comparative ink prints from same region of real finger and artificial finger from human subject. Note only portion of entire print is shown for data protection purposes.

**Figure 4 viruses-14-01048-f004:**
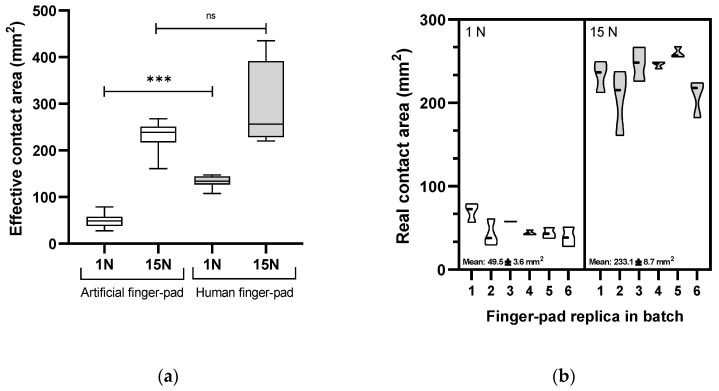
Characterization of artificial finger-pads. (**a**) Comparison of effective contact areas of artificial finger-pad and human finger counterpart at two forces, and (**b**) reproducibility of effective contact area across a batch of replica finger-pads. *** = significance (*p* < 0.05), ns = not significant (*p* > 0.05).

**Figure 5 viruses-14-01048-f005:**
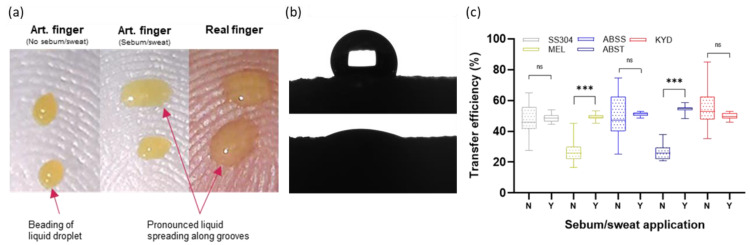
Impact of artificial eccrine sweat–sebum application. (**a**) Effect on surface wettability of artificial finger-pad compared to human finger, (**b**) cross-section photograph of artificial saliva on artificial finger-pad in absence and presence of eccrine sweat–sebum emulsion during measurement of surface energy, and (**c**) effect on transfer efficiency of liquid droplet to artificial finger-pad. *** = significance (*p* < 0.05), ns = not significant (*p* > 0.05).

**Figure 6 viruses-14-01048-f006:**
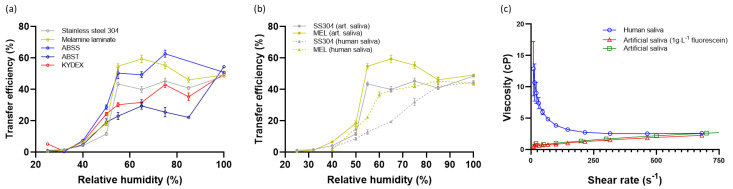
Effect of relative humidity on transfer from surfaces to artificial finger-pads. (**a**) Artificial saliva, (**b**) pooled human saliva, and (**c**) rheological properties of artificial and human saliva. Values are mean ± standard error. Note: the 100% values correspond to immediate transfer of a 1 µL deposited droplet (i.e., within 5 s of deposition).

**Figure 7 viruses-14-01048-f007:**
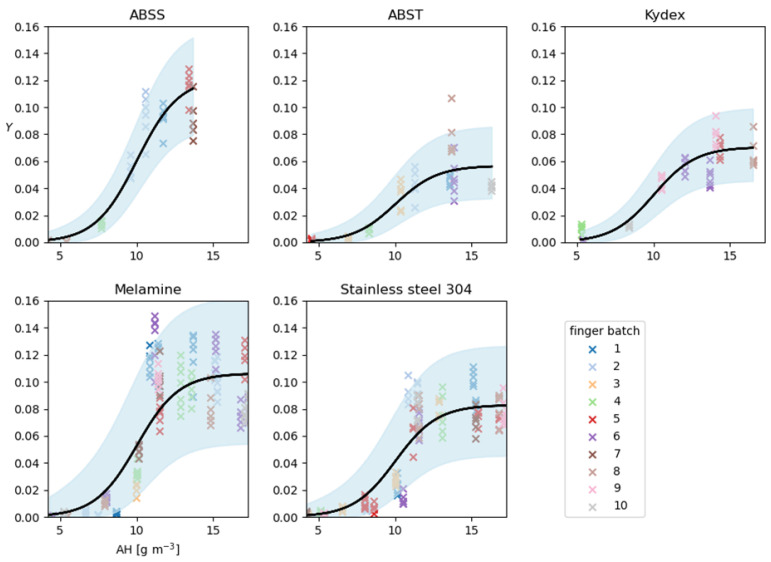
Posterior-predictive plot for the fitted non-linear regression model. Shaded region represents the 2.5 to 97.5 percentile range, dependent upon AH, for the posterior-predicted distribution for the replicate data Y and the black lines show the mean predicted Y. The model was fitted to the aerosol deposition data only.

**Figure 8 viruses-14-01048-f008:**
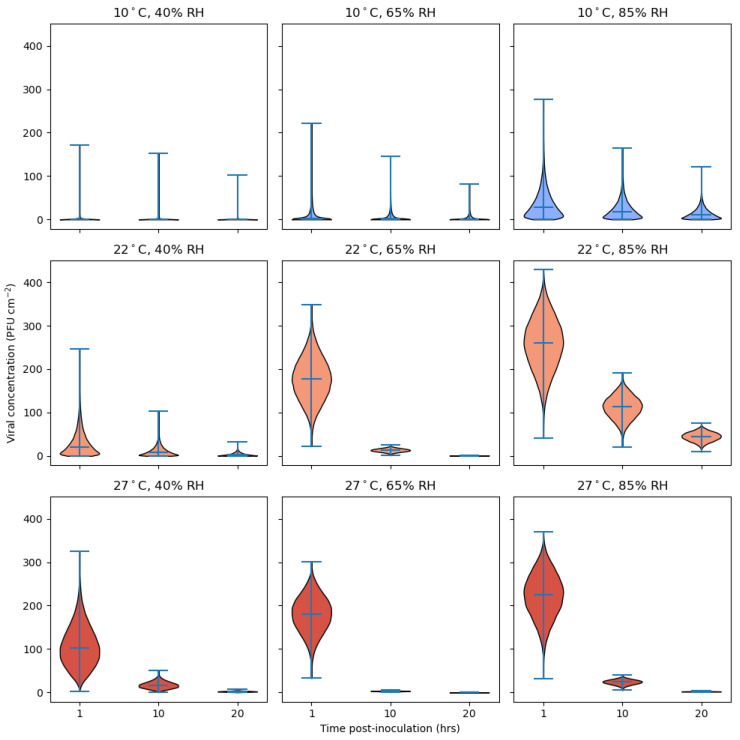
Estimates of viable SARS-CoV-2 transferred to human finger under different environmental conditions. The horizontal lines within each violin indicate the median viral concentration as well as the extrema. Transfer efficiencies were sampled from beta distributions obtained from the fitted regression model with parameters fixed at posterior median values for a melamine surface (see Appendix A).

**Table 1 viruses-14-01048-t001:** Surface roughness and wettability characteristics of coupon surfaces.

Coupon Surface	Example High-Touch Surfaces Represented	Roughness Profile Parameters ^1^			Surface Energy (mN·m^−1^) ^2^	Contact Angles ^3^	
		R_a_	R_q_	R_z_		θ_Adv_; θ_Rec_	Δθ_r_
**SS304**	Door handles, hand poles, keypads, taps, cutlery	0.167 ± 0.024	0.250 ± 0.046	4.05 ± 0.73	26.3 ± 0.26	80.6; 44.3	14.2
**ALU**	Door push plates, handrails, furniture handles	0.454 ± 0.037	0.639 ± 0.048	6.94 ± 1.34	25.8 ± 0.36	92.7; 35.3	23.2
**MEL**	Table surfaces, kitchen cabinets, decorative panels	0.419 ± 0.024	0.524 ± 0.038	3.57 ± 0.71	29.2 ± 0.52	93.9; 38.2	20.3
**ABSS ^4^**	Entry/exit press, keypads, sockets, housing, dashboard	0.127 ± 0.014	0.174 ± 0.017	1.73 ± 0.097	28.9 ± 0.23	72.2; 47.4	11.1
**ABST ^4^**	Entry/exit press, keypads, sockets, housing, dashboard	33.11 ± 0.67	38.87 ± 0.78	165.7 ± 10.8	33.7 ± 0.43	73.8; 32.1	26.4
**KYD ^4^**	Trays, storage lockers, hand rests, seat backs, holsters	4.49 ± 0.186	5.53 ± 0.245	30.3 ± 1.81	28.3 ± 0.32	82.5; 43.5	15.0
**Silicone ^5^**	No sweat–sebum Sweat–sebum	ND	ND	ND	26.7 ± 0.37 ND	82.6; 68.4 68.9; 58.5	ND
**Art. silicone finger-pad**	No sweat–sebum Sweat–sebum	ND	ND	ND	28.2 ± 0.34 56.2 ± 0.72		ND

^1^ Ra = arithmetical mean of assessed profile; Rq = root mean squared assessed profile; Rz = maximum peak to valley height of the profile; ^2^ = total surface energy; polar and dispersive components are provided in Appendix A; ^3^ Δθ_r_ = difference between the recipient (sebum-coated silicon covered SS304) and donor (coupon) surface receding contact angles (θ_Rec_); ^4^ = sample glued together with contacted textured surfaces facing external; ^5^ = silicon stub for surface energy or silicon-coated SS304 coupon. Note: contact angles are for artificial saliva. ND = not determined.

## Data Availability

Data and codes underlying the model fitting and QMRA are available at: https://github.com/leeds-indoor-air/Touch_transfer_experiment_data_analysis.git. Raw ABST and SS304 topology data are available at: DOI: 10.15131/shef.data.19738264.

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
