# Peer review of "Effect of Relative Humidity on Transfer of Aerosol-Deposited Artificial and Human Saliva from Surfaces to Artificial Finger-Pads"

_viruses, 2022, doi:10.3390/v14051048_

Round 1

Reviewer 1 Report

Walker et al. describe an experimental setup to quantitatively examine the touch transfer of aerosol deposited infectious material from different surfaces. The authors further perform a quantitative Microbial Risk Assessment (QMRA) to derive predictions about transfer efficiency of SARS-CoV-2 from surfaces to human skin, and how this is affected by environmental conditions.

Overall the paper seems of high quality and the experiments are performed at a high standard. The manuscript is recommended for publication. If possible, the authors might consider to validate the model using infectious SARS-CoV2 particles and/or a suitable surrogate virus. Moreover, the text, in particular the discussion section is very long (7+ pages), the authors might consider to shorten it.

Author Response

Dear Sir/Madam,

Thank you for the kind comments. Unfortunately the validation experiments with SARS-CoV-2 were outside scope of this piece of research, though we do have plans to initially progress to 100 nm beads and phi6 then to a pathogenic virus. They key for us is to ensure at each step we have good aerosol and humidity control. We did consider the length of discussion at time of writing and decided that inclusion and discussion of the limitations of the experimentation, modelling and existing literature data was important. We think promoting these limitations provides the whole research area direction for linking the hazard posed by respiratory aerosol and droplets to deposition on surfaces and subsequent factors that may influence surface transmission, and to date the connections seem largely overlooked. If you are content with these reasons, then I will leave discussion as it stands. Best wishes, Rich

Reviewer 2 Report

This manuscript presents very interesting research on how humidity affects the transmission of the pathogen between surfaces. Although I am not an expert in this field, the research design and results can support the author's conclusion. This research may provide insights to improve the design of surfaces frequently touched by the public to reduce the risk of contact transmission. 

Author Response

Dear Sir/Madam, Many thanks for your comments. It was an interesting area to conduct research, and i think supports the linkage and evolution of hazard from respiratory tract to aerosol/droplet, to surface deposition and contact hazard. Best wishes, Rich